# No evidence for causal effects of trust in science on intentions for health-related behavior
Tobias Wingen [1], Ann-Christin Posten [2] & Simone Dohle [3] ✉

Many researchers and policymakers assume that interventions targeting trust in science will be key for promoting health-related behaviors, including in the context of curbing the spread of disease. One central finding from the past pandemic is that trust in science predicted health-related protection intentions and behaviors, such as social distancing and vaccination. Yet, it is unclear whether the observed correlation between trust in science and protection intentions does indeed imply causation. Across our studies (total $N = 5311$), we successfully replicated this correlation. However, when experimentally manipulating self-reported trust in science, we found no evidence for causal effects on protection intentions. This absence of meaningful effects was confirmed by equivalence tests, an internal meta-analysis ($N = 3761$), and a machine learning algorithm. These results question the causal importance of short-term changes in trust in science for protection intentions. Drawing the right lessons from the COVID-19 pandemic will be essential for effective future policy responses.

What lessons can we learn from COVID-19-related research? Now, more than two years after the WHO chief declared the end to COVID-19 as a global health emergency[1], it is crucial for policymakers and researchers to identify insights to curb the spread of future pandemics. Indeed, researchers have started to systematically synthesize evidence for policy-makers, with the goal of providing tools for the next pandemic or other crises[2]. Especially the social and behavioral sciences can provide valuable insights by bridging the gap between scientific recommendations and public compliance[3]. For example, identifying psychological drivers of protection intentions and behaviors could provide relatively cost-efficient and practical tools to prevent infections, benefiting individuals and society alike.

One widely-reported factor associated with protection intentions and behavior is trust in sciencee.g.,[4–11]. For example, trust in science was one of the strongest predictors of protection intentions (e.g., intentions to engage in social distancing, hand hygiene, or mask-wearing) soon after the start of the pandemic in Germany[4]. Later work observed that trust in science predicted individuals' protection intentions across 23 countries[5] and that trust in science on a societal level was a relevant determinant of societies' resilience in their fight against the pandemic[6]. Regarding vaccines (one of the most effective measures against the COVID-19 pandemic[12]), trust in science also predicted vaccine confidence in general[7] and directly related to COVID-19[8]. Furthermore, recent work suggests that trust in science played an important mediating role, for example, in the link between spirituality and protection intentions[13].

Based on this broad evidence, it is a reasonable assumption that strengthening trust in science could be a key tool to increase protection intentions and tackle future pandemics. Likewise, it could be assumed that damaging trust in science (e.g., by raising unjustified doubts about science and scientists) could harm protection intentions. In other words, one possible interpretation is that this correlation implies causation and that trust in science thus causes protection intentions. In fact, many researchers and policymakers did take these correlational findings as a hint of a causal relationship, e.g.,[4]. For example, they described trust in science as a key driving force behind individual compliance with protective measures[6] or argued that enhancing trust in science would be crucial to promote behavioral change[5]. Likewise, the relevant chapter on this research in a report by *Eurofound* (an EU agency providing knowledge to improve policies) is literally titled "The influence of trust in complying with COVID-19 measures"[14, p.17], suggesting that this causal interpretation found its way to policymakers.

From a theoretical perspective, such a causal interpretation could be seen as reasonable. Trust is typically defined as a willingness to make oneself vulnerable to another in anticipation of beneficial outcomes[15,16]. In this sense, many recommended protective behaviors involve placing oneself in a vulnerable position by following scientists' advice (for example, by accepting a potentially harmful vaccine) with the expectation of a positive outcome, such as gaining immunity against infections. Therefore, individuals may be more likely to engage in scientifically recommended behaviors if their trust

[1]FernUniversität in Hagen, Hagen, Germany. [2]University of Limerick, Limerick, Ireland. [3]University of Bonn, Bonn, Germany. ✉e-mail: simone.dohle@ukbonn.de

in science is high. This also aligns with classic models of persuasion, which emphasize that messages from trustworthy sources are more persuasive[17,18].

At the same time, it is widely recognized that correlation does not logically imply causation[19–21] and could, for example, also result from (unmeasured) common causes. Therefore, previous research tried to control for various confounding factors, such as age, gender, or socioeconomic status[4–7]. However, as human cognition and behavior are highly complex, it is virtually impossible to prove that all sources of confounding have been controlled for[21]. Consequently, most of the correlational work emphasized this limitation, for example, by noting that "findings are based on cross-sectional data that prevents us from making causal claims. Thus, future research should confirm the reported insights by using experimental designs"[5, p.11].

This points to a problem: on the one hand, policymakers need to draw lessons from the pandemic to improve future policies; on the other hand, the evidence is limited, as it is primarily based on correlational data. Thus, there is an urgent need for experimental studies[22]. So far, to the best of our knowledge, only one proof-of-concept study has tried to manipulate trust in science in this pandemic context. However, the manipulation only had a very small effect on trust in science[22], and the observed null findings are thus difficult to interpret.

Given the stark evidence of correlational results linking trust to protection intentions and the lack of causal evidence regarding this relationship, we were motivated to investigate this causal role of trust. In line with previous literature, we initially hypothesized that trust in science has a causal and positive effect on all investigated protection intentions (social distancing, hand hygiene, mask-wearing, and vaccination). To ensure that we could detect a causal relationship between trust and protection intentions, we developed and pretested a stronger ($d = 0.74$) self-reported trust in science manipulation in the context of a pandemic.

It is, however, essential to recognize that trust in science can be conceptualized on multiple levels, such as trust in science as an institution, trust in scientists, or trust in scientific methods[23]. Our studies focused specifically on trust in scientists, as they are likely the most visible aspect of science to the public. Another important distinction is between trait (i.e., trust in science as a relatively stable personality disposition) and state-level components of trust (i.e., trust fluctuating with situational factors). Recent work suggests that, like most psychological constructs[24], trust encompasses state and trait components[25]. Given the brief and experimental nature of our intervention, we likely primarily influenced the situational, state-level trust in science. We return to this distinction in the discussion section.

## Methods
### General approach concerning all studies
We report all measures and conditions of interest to the studies' research question(s) in the manuscript (an overview of all translated measures is presented on the OSF). We report a prestudy, in which we pretested our trust in science materials in the Supplementary Methods. We do not report one study ($N = 260$) where our exclusion criteria were unintentionally confounded with the experimental condition, rendering the results uninterpretable (i.e., we can not tell whether observed effects would be due to imbalanced exclusions or our manipulation). All participants who completed our studies were included in the analyses, except if they met preregistered exclusion criteria. In all studies, our experimental groups consisted of distinct samples, and no repeated measures designs were employed. All studies were preregistered, which included information about the research question, dependent variables, conditions, analyses, sample sizes, and outlier handling. It nevertheless has to be noted that we expected causal effects of trust in science in most studies and preregistered a variety of analyses and hypotheses to follow up on these effects (e.g., mediation analyses, comparisons involving the "no manipulation" control groups to understand the direction of the effects). As we eventually did not observe such effects, we do not report these (now pointless) follow-up analyses here and instead focus only on the direct effects. Moreover, we did not foresee the need to provide evidence for null effects. Thus, analyses focusing on null

effects (e.g., the internal meta-analysis, the machine learning approach, and the equivalence tests) were not preregistered. A detailed account of what was preregistered for each study is provided through the preregistration documents on the OSF. The Ethics Commission of the Faculty of Human Sciences, University of Cologne (Protocol ID: SDHF0105) approved this research project. Participants provided informed consent before participation.

Statistical analyses were conducted using $R^{26}$, and for the primary analyses, we relied on the packages *lavaan*[27], *metafor*[28], *pwr*[29], *psych*[30], *ranger*[31], *tidymodels*[32], *tidyverse*[33], and *TOSTER*[34,35]. All statistical tests were two-sided unless stated otherwise. For null hypothesis significance testing (NHST), we used regular *t*-tests assuming equal variances. However, for equivalence tests, we used Welch's *t*-test, as it is the default in the *TOSTER* package. This decision was not based on formal assumption tests, as our large sample sizes are generally robust to violations of normality and variance homogeneity[36]. No covariates were included in these analyses. Portions of the code for these analyses were co-developed using generative AI (ChatGPT-4o), but all code was ultimately tested and validated by the research team.

### Study 1
As an initial study, we aimed to replicate prior correlational findings showing that trust in science is a central predictor of intentions for protective health behavior during a pandemic. Specifically, we sought to replicate one of the earliest studies finding this link[4], published in December 2020. We aimed to establish that this finding was replicable, even later in the pandemic. Moreover, we extended previous research by investigating vaccination intentions, which had not been assessed earlier due to the unavailability of vaccines at the time.

**Participants and design.** To replicate the correlation between trust in science and protective intention, we conducted a cross-sectional online study in Germany after the first two COVID-19 waves (March 2021). Participants were recruited through an ISO-certified (ISO 20252:2019) internet panel provider (Respondi AG, Germany), for a compensation of €1.00. The sample was representative of the German population in terms of gender and age. The inclusion criterion was that the participants had not yet been vaccinated against COVID-19, as we also measured vaccination intentions. Note that at the time of the study, vaccination was only available for high-risk groups in Germany (only 3.6% of the German population were fully vaccinated at this time[37]). After providing informed consent, participants were provided with all measures. The final sample (after excluding 102 participants who failed a preregistered attention and conscientiousness check) consisted of 326 participants ($M_{age} = 44.4$, $SD_{age} = 14.6$, 162 women, 162 men, 2 non-binary individuals). Power analyses revealed that the sample size of 326 had a 99.99% power to detect a correlation similar to the one between trust in science and acceptance of protective measures ($r = 0.34$) observed by Dohle and colleagues[4] (2020, Study 1).

**Procedure and measures.** Following Dohle and colleagues[4] (2020), trust in science was measured using five items (e.g., "I trust German scientists to do what is right during the Corona crisis"; α = 0.92; adapted initially from Nisbet and colleagues[38]). For all measures, participants indicated their agreement on a 7-point Likert scale ranging from 1 ("strongly disagree") to 7 ("strongly agree"). We measured intentions to engage in protective behavior using separate items for each behavior. We measured physical distancing intentions using five items (e.g., "In the next four weeks, I intend to always keep a distance of 1.5 m in public transport", α = 0.95, note that 1.5 m were the recommended distance in Germany at that time), hand hygiene intentions using four items (e.g., "In the next four weeks, I intend to wash my hands thoroughly after coming home," α = 0.90), and mask-wearing-intentions using four items (e.g., "In the next four weeks, I intend to wear a protective medical mask in supermarkets and other shops," α = 0.91). For vaccination, four items

were used to measure intention (e.g., "If I were offered a vaccination appointment in the next week where I would be vaccinated with the BioNTech/Pfizer vaccine, I would attend," α = 0.93), representing the four available vaccines at that time. To ensure data quality, we collected a preregistered data quality check ("If you read this, choose 1") and a conscientiousness check ("In your honest opinion, should we use your data in our analyses in this study")[39]. Finally, we collected socio-demographic characteristics including, among others, gender, age, income, education, and occupation[40] (see OSF materials for the original and translated wording of items). We did not collect data on race or ethnicity in any of our studies.

## Study 2

Study 2 tested whether these correlations between trust in science and protection intentions reflect causal effects. We thus experimentally manipulated trust in science by presenting pretested, actual statements during the COVID-19 pandemic. We selected statements from the same scientists, which later turned out to be primarily correct (high trust condition) or incorrect (low trust condition). Details on the pretest, where we found that this manipulation affects self-reported trust in science both in *general*, $t(112) = 3.90$, $p < 0.001$, $d = 0.74$, 95%-CI[0.36, 1.12], and in its different facets *benevolence* ($t(112) = 2.63$, $p = 0.010$, $d = 0.50$, 95%-CI[0.12, 0.87], *expertise*, $t(112) = 3.34$, $p = 0.001$, $d = 0.63$, 95%-CI[0.25, 1.01]), and *integrity*, $t(112) = 3.67$, $p < 0.001$, $d = 0.69$, 95%-CI[0.31, 1.07]), can be found in the Supplementary Methods.

**Participants and design**. In this experimental online study, we again used quota-sampling (for age and gender) and recruited 18–59-year-old participants via the same online panel provider as before, for a compensation of €1.00. Only individuals who had not received a COVID-19 vaccine yet could participate (inclusion criterion). Note that older participants (60 + ) were not included in this study because they were prioritized in the German vaccination campaign, and we assumed that most people in this age group who intended to be vaccinated had already been vaccinated. The study was conducted in May 2021 (9.4% of the German population were fully vaccinated at this time[41]).

Participants first completed a trust in science manipulation developed in a prestudy (see Supplementary Methods for details). Previous work already aimed to experimentally investigate the effects of trust in science on protection intentions, but the manipulation used had a minimal and, depending on the analyses, statistically non-significant effect on trust in science, making any causal inferences difficult[22].

For our trust in science manipulation, we build on prior research, which found that trust in science can be manipulated by providing information about the percentage of reliable research results[23]. We, therefore, developed a similar trust in scientists manipulation for the pandemic context. Participants read authentic statements made by real scientists related to COVID-19 (well-known biologists, medical scientists, and virologists). Crucially, depending on their condition, we presented statements by the same scientists that later turned out to be largely incorrect (2 out of 10, low-trust condition) or mostly correct (8 out of 10, high-trust condition). After each statement, participants learned whether that statement turned out correct or incorrect. The statements (see OSF materials for for the original and translated wording) focused on expected epidemiological trends (e.g., waves, seasonality), typical symptoms of an infection, the dangerousness of COVID-19, and its transmissibility. Notably, we included **no** statements directly related to protective behaviors (e.g., on the effectiveness of mask-wearing or similar). In a prestudy, this manipulation strongly ($d = 0.74$) affected general self-reported trust in science and different facets of trust (see Supplementary Methods for details).

Nevertheless, it is important to note that the manipulation solely focused on predictions made by COVID-19 scientists. In contrast, it did not address trust in science more broadly, nor aspects such as scientific methods or the scientific process, which inherently involve uncertainty and knowledge updating[42].

The final sample (after excluding 277 participants who failed pre-registered data quality, attention, and/or conscientiousness checks or fell outside the preregistered age range) consisted of 1707 participants ($M_{age} = 40.0$, $SD_{age} = 12.2$, 881 women, 818 men, 8 non-binary individuals), again recruited using a panel provider (Respondi). Power analyses revealed that our final sample ($n = 861$ in the low trust condition, $n = 846$ in the high trust condition) had a very large power ($1 - \beta = 98.5\%$) to detect our smallest effect size of interest. Please note that for this power analysis and for our equivalence tests, we (post-hoc) defined $d = 0.2$ as the smallest effect size of interest, as $d = 0.2$ is conventionally considered a small effect[43].

**Procedure and measures**. Participants completed the trust in science manipulation developed in the prestudy and thus read actual scientific statements about the pandemic (see Supplementary Methods for details) that turned out to be mostly correct (high trust condition) or incorrect (low trust condition). As a manipulation check, we then measured participants' self-reported trust in science using the same five items as in Study 1 (α =0.93). As a dependent variable, we then measured intentions to engage in protective behavior using the same items and types of behavior as in Study 2: physical distancing (α = 0.94), hand hygiene (α = 0.88), mask-wearing (α = 0.88), and vaccination intentions (α = 0.90, note that we report the α across all vaccination questions, while our analyses only focus on intentions related to mRNA vaccines, as pre-registered). We utilized the same quality and conscientiousness checks as in Study 1.

We finally added an attention check at the end of the study to ensure that potential null results were not due to a lack of attention. As an attention check, participants (re)read six statements from scientists and indicated for each statement whether they had seen it as part of the study materials, had not seen it, or did not know. Three statements were part of the low trust condition, and three were part of the high trust condition, which balanced the amount of known statements in both conditions. Participants who answered too many questions incorrectly (see OSF-preregistration for details) were excluded from the analyses.

Finally, we collected sociodemographic characteristics, including, among others, gender, age, income, education, and occupation.

## Study 3

Study 2 found no evidence that the positive correlation between self-reported trust in science and protection intentions reflects a causal link. We did not only observe null effects but even adverse (small) effects of trust in science on vaccination and mask-wearing intentions. One reason for these unexpected results could be that our intervention may unintendedly have affected other variables that may also affect the dependent variables. This problem is widespread when targeting broad psychological constructs such as trust in science[21], which are difficult to manipulate without manipulating related constructs.

Indeed, we suspected that our trust in science manipulation, which described incorrect statements by scientists, could also have affected fear. We assumed that, by indicating that the pandemic is beyond scientific understanding, we may have induced increased fear in the low trust condition. As fear is associated with increased protection intentions[44] and thus may work in the opposite direction as trust in science, this could explain why we observed no or even adverse effects of our trust in science manipulation on protection intentions.

Moreover, it seemed possible that sample differences (e.g., time, which seems to play a particular role given the consistent developments in the ongoing pandemic at the time of study conduct) may have caused the different results between earlier research and Study 2. We thus included a "no manipulation" condition in Study 3, in which we again aimed to replicate the correlational findings of earlier research.

**Participants and design**. Again, we recruited a sample of the German population regarding gender and age using a panel provider (Respondi with a compensation of €1.00) and conducted an experimental online

study. We used quota-sampling for age and gender, and recruited 18–59-year-old participants via the same online panel provider as before. As an inclusion criterion, only individuals who had not received a COVID-19 vaccine yet could participate. The study was conducted in June and July 2021, when 26.2% of the German population was fully vaccinated[45].

Participants completed the same trust manipulation as in Study 2, except that we also employed a condition that received "no manipulation," which we used to conduct correlational analyses. After excluding 386 participants who failed preregistered data quality, attention, and/or conscientiousness checks, our final sample consisted of 2,347 participants ($M_{age}$ = 40.6, $SD_{age}$ = 12.3, 1,326 women, 1,015 men, 4 non-binary individuals). Power analyses revealed that our final sample ($n$ = 726 in the low-trust condition, $n$ = 715 in the high-trust condition, $n$ = 906 in the "no manipulation" condition) again had a very large power ($1 - \beta$ = 96.7%) to detect our smallest effect size of interest ($d$ = 0.2).

**Procedure and measures.** Participants completed the trust in science manipulation developed in Study 2 and thus read correct and incorrect scientific statements about the pandemic (see Study 2 for details). As a manipulation check, we then measured participants' self-reported trust in science using the same five items as in Studies 1 and 2 ($\alpha$ = 0.93). As a potential (additional) mediator, we measured COVID-19-related fear with three items (e.g., "The novel coronavirus is …") on a 7-point Likert scale ranging from 1 ("nonscary") to 7 ("scary"). As a dependent variable, we then measured intentions to engage in protective behavior using the same items and types of behavior as in Studies 1 and 2: physical distancing ($\alpha$ = 0.95), hand hygiene ($\alpha$ = 0.91), mask-wearing ($\alpha$ = 0.92), and vaccination intentions ($\alpha$ = 0.90, note that we again report the $\alpha$ across all vaccination questions, while our analyses only focus on intentions related to mRNA vaccines, as preregistered). We also exploratorily assessed pandemic fatigue[46], mood[47], and perceived vulnerability and perceived severity of COVID-19 (see OSF materials for the original and translated wording of items). We utilized the same data quality, conscientiousness, and attention checks as in Study 2.

Finally, we collected sociodemographic characteristics, including, among others, gender, age, income, education, and occupation (see OSF materials for details).

## Study 4

Study 3 again demonstrated a striking discrepancy between the correlational and causal effects of trust in science on protective intentions. While we observed strong correlations between trust in science and protection intentions, we found no evidence that manipulating trust in science causes protection intentions. We further ruled out various (potential) confounds of our trust in science manipulation, particularly fear.

It thus seemed possible that behavioral intentions in the (then) ongoing pandemic were already too strong and established to be swayed by our trust in science manipulation. To test this idea, we assessed the effects on protection intentions in a fictional future pandemic. We described this pandemic as occurring ten years in the future, involving a novel, dangerous virus similar to COVID-19, in which scientists thus recommend the same protective measures. Due to the fictional and distant nature of this pandemic, we expected that aspects like fear would be less relevant, allowing us to finally detect the causal effects of trust in science. As in Study 3, we again included a "no manipulation" condition, in which we aimed to replicate the correlational findings of Studies 1 and 3.

**Participants and design.** We recruited a sample using Prolific (with a compensation of £1.88) and conducted an experimental online study. A quota was set for gender (50% female/male), and Prolific was set up to target only participants located in Germany with German as their first language. The study was conducted in January 2022.

Participants first completed the same trust in science manipulation as in the previous studies. The final sample (after excluding 35 participants who failed preregistered data quality, attention, and/or conscientiousness checks) consisted of 931 participants ($M_{age}$ = 28.7, $SD_{age}$ = 8.6, 479 women, 437 men, 15 non-binary individuals). Power analyses revealed that our final sample ($n$ = 308 in the low-trust condition, $n$ = 305 in the high-trust condition, $n$ = 318 in the no-manipulation condition) had a 69.6% power to detect our smallest effect size of interest ($d$ = 0.2).

**Procedure and measures.** Participants completed the same trust in science manipulation as in Study 3, including the third "no manipulation" condition. As a manipulation check, we then measured participants' self-reported trust in science using the same five items as in Study 1 ($\alpha$ = 0.91). As a dependent variable, we again assessed intentions to engage in protective behavior, but given the strong reliabilities observed in the earlier studies and to keep the study concise, we just used one item for each construct. We utilized the same attention, conscientiousness, and data quality checks as in Study 1.

Finally, we collected sociodemographic characteristics, including, among others, gender, age, income, education, and occupation.

## Combined analyses

Overall, we observed no evidence that trust in science has positive causal effects on protective intentions. However, it seemed possible that these effects could emerge when meta-analytically considering the combined data across all studies, resulting in a higher statistical power. To test this idea, we conducted an internal meta-analysis and aggregated evidence across the relevant Studies 2 to 4.

Moreover, we considered the possibility that trust in science may not have a simple but a rather complex causal effect on protection intentions. For example, the effect could be moderated by other variables, such as age, gender, or political orientation, or by even more complex interaction patterns. In this case, trust in science would still have a causal, yet very complex, effect on protection intentions. Our simple statistical analyses (i.e., $t$-tests) would miss such complex relations, but certain machine learning algorithms could potentially reveal them. We thus tested whether including manipulated trust in science as a predictor would improve the performance of a random forest machine learning algorithm, which is capable of modeling highly complex effects, such as interactions and non-linear relationships[48].

## Internal meta-analysis

To estimate the causal effects of trust in science across studies, we reanalyzed the data of Studies 2 to 4 (total $N$ = 4985; 3761 without the "no manipulation" condition) and conducted a random effect meta-analysis using the *metafor* package in $R$[28] for each dependent variable: physical distancing, hand hygiene, mask-wearing, vaccination, and finally also for trust as a manipulation check.

## Machine learning approach

Finally, we considered the possibility that trust in science might not have a simple, direct causal effect on protection intentions. Instead, it seemed possible that the causal link between trust in science and protection intentions is more complex and, for instance, moderated by other variables, such as age, gender, political orientation, or even more complex interaction patterns. In this case, trust in science would still have a causal, yet very complex, effect on protection intentions. Such complex relations, however, would be missed by many classic statistical methods, such as $t$-tests, but can be revealed by more advanced analyses. To avoid hypothesizing post hoc and selectively testing only specific moderations (e.g., in linear regressions), we instead employed a machine learning algorithm to explore potential interaction patterns more broadly.

Some of the best "off-the-shelf" machine learning models are random forests[49], which are capable of modeling highly complex interactions and non-linear relationships[48]. Such models are typically evaluated by their predictive performance (e.g., cross-validated $R^2$), and including relevant variables (typically called *features* in the machine learning literature) as predictors in these models should boost their performance[50]. Thus, if there

are indeed (complex) effects of manipulated trust in science on protection intentions, including trust in science into the model should have a visible impact on its predictive performance.

To test this idea, we used data from Studies 2 to 4 and built predictive machine learning models incorporating participants' age, gender, income, education, occupation, political orientation, health, and COVID-19-related risk perception (for oneself and close others). All measures are presented in detail in the OSF. Using these features, we predicted each of the four protection intentions using the R package ranger[31] combined with tidymodels[32]. Critically, we ran each model twice: once including manipulated trust in science and once without it. We then compare the performance of both variants, indicated as 10-fold cross-validated $R^2$ values.

We finally repeated the same procedure using measured (instead of manipulated) trust in science to test whether including measured trust in science improves predictive performance. For these analyses, we used data from the control groups in Studies 3 and 4, where trust in science was measured but not manipulated.

### Reporting summary
Further information on research design is available in the Nature Portfolio Reporting Summary linked to this article.

## Results
### Study 1
Trust in science statistically significantly and strongly predicted all four types of investigated protection intentions. It correlated with intentions related to *physical distancing*, $r(324) = 0.52$, $p < 0.001$, 95%-CI[0.44, 0.60], *hand hygiene*, $r(324) = 0.34$, $p < 0.001$, 95%-CI[0.24, 0.43]), *mask-wearing*, $r(324) = 0.49$, $p < 0.001$, 95%-CI[0.40, 0.57]), and most strongly with intentions related to *vaccination*, $r(324) = 0.58$, $p < 0.001$, 95%-CI[0.51, 0.65]. These patterns are depicted in Fig. 1.

### Study 2
Study 2 moved beyond correlation by experimentally manipulating trust in science to test its causal effect on protective intentions. We observed that our trust in science manipulation was successful. Participants in the high trust

condition showed higher self-reported trust in science ($M = 5.0$, $SD = 1.5$) than participants in the low trust condition ($M = 4.3$, $SD = 1.4$), $t(1705) = 10.19$, $p < 0.001$, $d = 0.49$, 95%-CI[0.40, 0.59], again indicating a substantial effect of our manipulation.

Contrary to our expectations, our trust in science manipulation, however, did not statistically significantly affect participants' intentions to engage in *physical distancing*, $t(1705) = 0.64$, $p = 0.524$, $d = 0.03$, 95%-CI[-0.13, 0.06], nor to increase *hand hygiene*, $t(1705) = 0.22$, $p = 0.828$, $d = 0.01$, 95%-CI[-0.08, 0.11].

Unexpectedly, we further observed that the high trust condition reported a statistically significantly lower intention to *wear masks* ($M = 5.8$, $SD = 1.5$), $t(1705) = 2.29$, $p = 0.022$, $d = 0.11$, 95%-CI[0.02, 0.21], compared to participants in the low trust-condition ($M = 6.0$, $SD = 1.4$). Moreover, participants in the high trust condition reported reduced *vaccination intentions* ($M = 4.9$, $SD = 2.4$), compared to participants in the low trust condition ($M = 5.3$, $SD = 2.2$), $t(1705) = 3.48$, $p < 0.001$, $d = 0.17$, 95%-CI[0.07, 0.26].

Using equivalence tests, we tested whether the observed effects are equivalent with an interval containing only very small effects ($d < 0.2$). This was the case for physical distancing $t(1698) = 3.49$, $p < 0.001$, hand hygiene, $t(1705) = 3.91$, $p < 0.001$, and mask-wearing, $t(1684) = -1.84$, $p = 0.033$. This was not the case for intentions to get vaccinated, $t(1687) = 0.68$, $p = 0.247$, which, however, was negatively influenced by trust in science, and the effect was thus also statistically significantly smaller than a positive, meaningful effect, $t(1687) = 7.57$, $p < 0.001$.

### Study 3
Study 3 replicated the design of Study 2 while more carefully considering potential confounds, particularly fear. In addition, it included a "no-manipulation" control condition, in which we again aimed to replicate the correlational findings of earlier research.

**Causal effects**. Our manipulation was successful. Participants in the high trust condition showed higher self-reported trust in science ($M = 4.5$, $SD = 1.6$) than participants in the low trust condition ($M = 3.8$, $SD = 1.5$), $t(1439) = 8.95$, $p < 0.001$, $d = 0.47$, 95%-CI[0.37, 0.58].

**Fig. 1 | Correlation between trust in science and various protection intentions.** The figure ($n = 326$ participants) shows scatter plots with linear regression lines and shaded areas around the regression lines, representing the 95% confidence intervals. The figure uses random jittering along both axes (± 0.15 units) to enhance the visibility of overlapping data points.

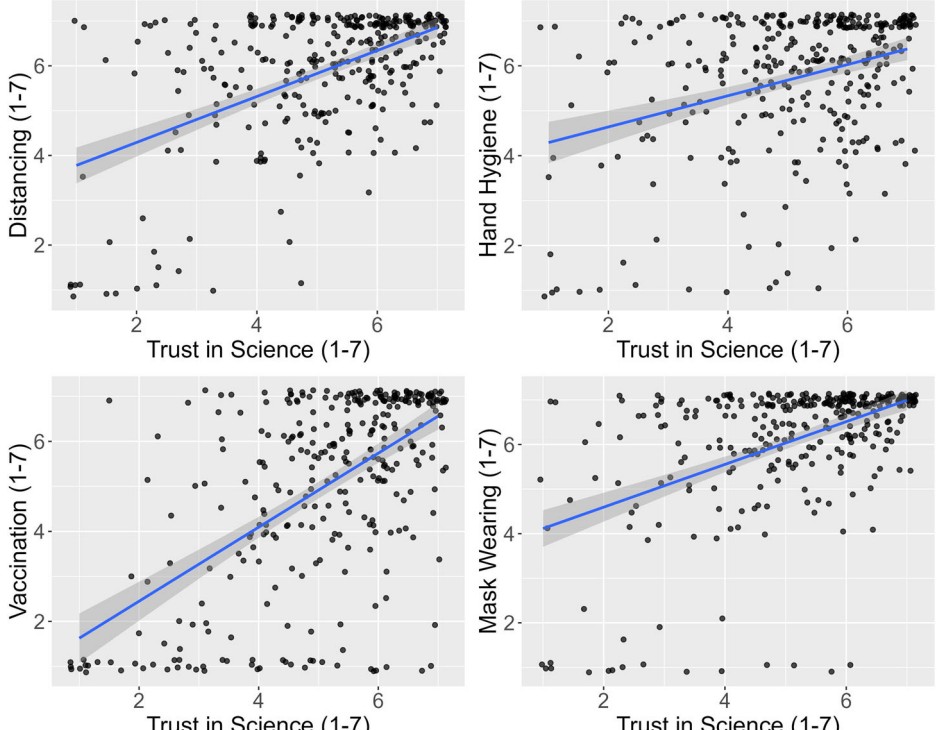

However, contrary to our expectation, we did not observe a statistical significant effect of our trust in science manipulation on fear, $t(1439) = 0.80$, $p = 0.422$, $d = 0.04$, 95%-CI[−0.06, 0.15]. Moreover, our manipulation also had no statistical significant effects on other potentially affected confounds. None of the dimensions of mood were statistically significantly affected by the trust-in-science manipulation. Specifically, the manipulation did not statistically significantly affect the dimension awake-tired, $t(1439) = 1.17$, $p = 0.241$, $d = 0.06$, 95% CI [−0.04, 0.17]; nor calm-nervous, $t(1439) = 1.91$, $p = 0.056$, $d = 0.10$, 95% CI [0.00, 0.20]; nor good-bad, $t(1439) = 1.07$, $p = 0.287$, $d = 0.06$, 95% CI [−0.05, 0.16].

The trust-in-science manipulation also did not statistically significantly affect pandemic fatigue, $t(1439) = 1.13$, $p = .257$, $d = 0.06$, 95% CI [−0.04, 0.16]; perceived vulnerability to COVID-19, $t(1439) = 0.46$, $p = .644$, $d = 0.02$, 95% CI [−0.08, 0.13]; or perceived severity of COVID-19, $t(1439) = 1.04$, $p = 0.299$, $d = 0.05$, 95% CI [−0.05, 0.16].

Moreover, our trust in science-manipulation did not statistically significantly affect intentions to engage in *physical distancing*, $t(1439) = 1.22$, $p = 0.222$, d = 0.06, 95%-CI[−0.04, 0.17], nor to increase *hand hygiene*, $t(1439) = 1.31$, $p = 0.190$, $d = 0.07$, 95%-CI[−0.03, 0.17], nor to *getting vaccinated*, $t(1439) = 0.12$, $p = .905$, $d = 0.01$, 95%-CI[−0.10, 0.11] and also not *mask-wearing intentions* $t(1439) = 1.15$, $p = 0.250$, $d = 0.06$, 95%-CI[−0.04, 0.16].

Using equivalence tests, we again tested whether the observed effects are equivalent with an interval containing only very small effects ($d < 0.2$). This was the case for all investigated protection intentions: For physical distancing $t(1432) = 2.58$, $p = 0.005$, for hand hygiene $t(1432) = 2.48$, $p = 0.007$, for vaccination intentions, $t(1439) = 3.68$, $p < 0.001$, and for mask-wearing intentions, $t(1439) = 2.64$, $p < 0.004$.

**Correlational effects**. Using data from the "no manipulation" condition (which received no trust in science manipulation), we investigated whether the previously observed correlational effects of trust in science on protection intentions would be present in our sample. Indeed, in contrast to the (absent) causal effects described before, we observed substantial correlations of measured trust in science with intentions related to *physical distancing*, $r(904) = 0.48$, $p < 0.001$, 95%-CI[0.43, 0.53], *hand hygiene*, $r(904) = 0.34$, $p < 0.001$, 95%-CI[0.29, 0.40]), *mask-wearing*, $r(904) = 0.50$, $p < 0.001$, 95%-CI[0.45, 0.54]), and *vaccination*, $r(904) = 0.61$, $p < 0.001$, 95%-CI[0.57, 0.65].

### Study 4
Study 4 replicated Studies 2 and 3 using a fictional future pandemic scenario and again included a "no-manipulation" control condition.

**Causal effects**. Our manipulation was again successful. Participants in the high trust condition reported higher trust in science ($M = 5.9$, $SD = 1.0$) than participants in the low trust condition ($M = 5.0$, $SD = 1.1$), $t(611) = 10.74$, $p < 0.001$, $d = 0.87$, 95%-CI[0.70, 1.03].

Contrary to our expectations, our trust in science-manipulation did not statistically significantly affect intentions to engage in *physical distancing*, $t(611) = 1.09$, $p = 0.277$, $d = 0.09$, 95%-CI[−0.07, 0.25], nor to increase *hand hygiene*, $t(611) = 0.97$, $p = 0.334$, $d = 0.08$, 95%-CI[−0.08, 0.24], nor to *getting vaccinated*, $t(611) = 0.198$, $p = 0.843$, $d = 0.02$, 95%-CI[−0.14, 0.17], nor to *wearing a mask* in a future pandemic, $t(611) = 0.66$, $p = 0.507$, $d = 0.05$, 95%-CI[−0.10, 0.21].

Using equivalence tests, we tested whether the observed effects are equivalent with an interval containing only very small effects ($d < 0.2$). This was the case for getting vaccinated $t(610) = 2.28$, $p = 0.012$ and for mask-wearing $t(611) = 1.81$, $p = 0.035$, but not for physical distancing $t(596) = 1.39$, $p = 0.083$, and hand hygiene, $t(597) = 1.51$, $p = 0.066$. However, it is important to consider the reduced power of Study 4, which limits the interpretability of these non-significant results.

**Correlational effects**. As in Study 3, we again used the "no manipulation" condition to investigate whether we could find correlational effects

of trust in science on protection intentions, this time in a future pandemic. Indeed, in contrast to the (absent) causal effects described before, we observed substantial correlations of measured trust in science with intentions related to *physical distancing*, $r(316) = 0.41$, $p < 0.001$, 95%-CI[0.31, 0.50], *hand hygiene*, $r(316) = 0.25$, $p < 0.001$, 95%-CI[0.15, 0.35]), *mask-wearing*, $r(316) = 0.43$, $p < 0.001$, 95%-CI[0.34, 0.52]), and *vaccination*, $r(316) = 0.67$, $p < 0.001$, 95%-CI[0.61, 0.73].

### Combined analyses
Finally, we conducted two combined analyses across studies. First, we conducted a meta-analysis to enhance statistical power and evaluate the overall effects across all studies. Second, we conducted a machine learning analysis to explore potentially complex effects of trust in science on protective intentions.

### Internal meta-analysis
Our internal meta-analysis ($N = 3761$) again revealed that our manipulation was successful and showed a combined statistical significant effect of our manipulation on trust in science, $\beta = 0.60$, $SE = 0.12$, $p < 0.001$, 95%-CI[0.36, 0.85]. However, it revealed no statistical significant effect on any of the protection intentions, neither for *social distancing*, $\beta = −0.007$, $SE = 0.04$, $p = 0.871$, 95%-CI[−0.09, 0.07], nor for *hand hygiene*, $\beta = 0.007$, $SE = 0.04$, $p = 0.845$, 95%-CI[−0.07, 0.08], nor for *mask-wearing*, $\beta = −0.006$, $SE = 0.06$, $p = 0.923$, 95%-CI[−0.12, 0.11], and finally not for *vaccination*, $\beta = −0.056$, $SE = 0.063$, $p = 0.373$, 95%-CI[−0.18, 0.07]. This pattern is depicted in Fig. 2.

### Machine learning approach
Likewise, we observed no improved predictive performance of our random forest models after including *manipulated* trust in science as a predictor (see the left panel of Fig. 3). Thus, the algorithm did not reveal any evidence for a (complex) relationship between trust in science and protection intentions that could improve its predictive performance.

Including *measured* trust in science, however, led to an enormously improved predictive performance (see the right panel of Fig. 3). These findings thus mirror our earlier results that measured trust in science predicts protection intentions but that there is no statistical significant causal effect of manipulated trust in science, even when allowing for highly complex patterns and including various controls and potential interactions into the model.

### General discussion
Trust in science has been consistently identified as a central predictor of protection intentions during a pandemic e.g.,[4-11]. However, while we successfully replicated this correlation (Studies 1, 3, and 4), we observed no evidence that this correlation implies causation. Despite successfully altering people's self-reported trust in science and using high-powered designs, we observed no statistical significant positive effects on various protection intentions (Studies 2 – 4) and sometimes even unexpected negative effects (for mask-wearing and vaccination, Study 2). In most cases (except for vaccination in Study 2 and physical distancing/hand hygiene in Study 4), equivalence tests confirmed the absence of meaningful effects. These null effects were not due to differences in fear or other potential confounds (Study 3) and were also present when predicting protection intentions in a future pandemic (Study 4).

Likewise, combined analyses across all studies found no evidence for statistical significant causal effects of trust in science on protection intentions. An internal meta-analysis across Studies 2 to 4, which experimentally varied trust in science, observed null effects on all four protection intentions. The strongest positive effect (on hand hygiene) had a very small effect size and was stastically non-significant, despite a high statistical power to detect even very small effects, given the large combined sample of 3761 participants. Moreover, all meta-analytic confidence intervals were far below our criterion for potentially meaningful effects.

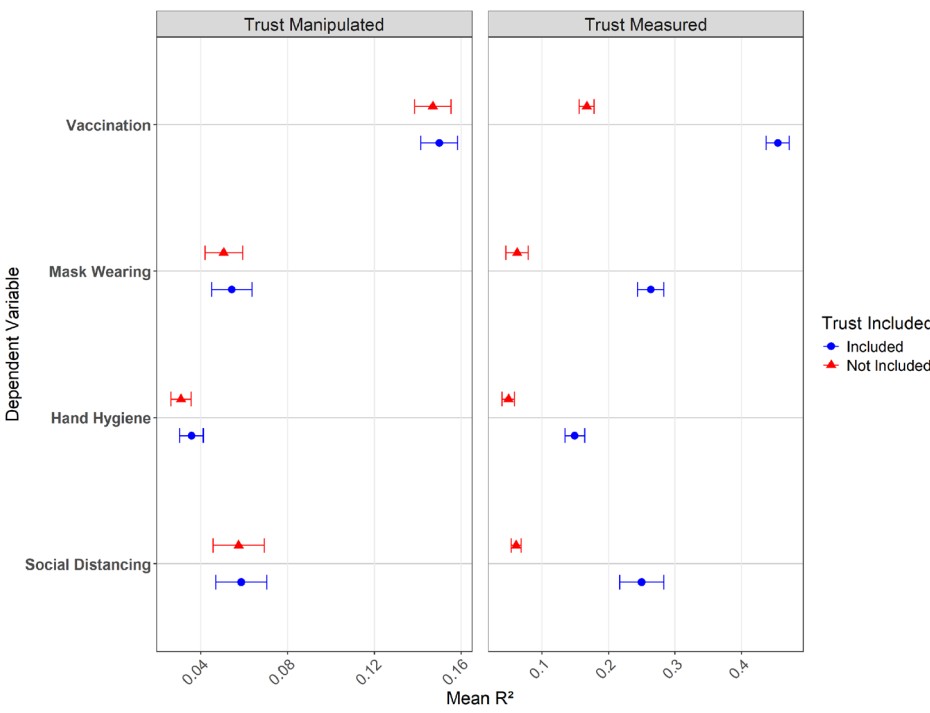

**Fig. 2 | Forest plot displaying the meta-analyic effects of manipulated trust in science on measured trust in science (manipulation check) and various protection intentions.** Error bars represent 95% confidence intervals around the estimated effect (with $n = 3761$ participants). The red dashed line represents an effect size of 0.

**Fig. 3 | Effect of including manipulated or measures trust in science on the predictive performance of a machine learning model.** Error bars represent the standard error of the mean $R^2$ values obtained through 10-fold cross-validation. Analyses for the left ($n = 3738$ participants) and right panes ($n = 1213$ participants) use different datasets.

Similarly, the performance of a machine learning algorithm did not statistically significantly improve after including participants' condition (i.e., manipulated trust in science) in the model. Random forest algorithms can generally model complex relationships (such as complex interaction patterns between trust in science and other variables). However, our results suggest that such patterns likely do not exist. Importantly, adding measured trust in science to the model dramatically improved the algorithm's performance. This again highlights the dichotomy between trust in science as an essential predictor but not as a causal driver of protection intentions.

## Limitations

Nevertheless, proving the absence of an effect is inherently difficult[35,51], and it is essential to consider alternative explanations and limitations. First, an effect may exist, but we failed to observe it because it is too small, our measures are too noisy, or our sample size is insufficient. However, our pretested trust-in-science manipulation demonstrated a large effect size in the pretest and medium-to-large effects in subsequent manipulation checks. Moreover, our measures were highly reliable across all investigated protection intentions. Finally, while we already used large sample sizes in the individual studies, we combined the data into an internal meta-analysis with 3761 participants, enabling the detection of even smaller effects. Nevertheless, it needs to be acknowledged that Study 4 may have had insufficient power on its own to rule out small but potentially meaningful effects.

Another explanation for the absence of effects could be that while our manipulation may have changed self-reported trust in science, it may not have changed all relevant dimensions of this trust. Based on classic theorizing on trustworthiness[52], trust in science and scientists is often conceptualized as comprising three dimensions: perceptions of scientists' benevolence, expertise, and integrity[15]. It seems possible that only certain dimensions of trust are relevant to protection intentions. However, although we only included a general measure of trust in science as a manipulation check, our pre-study revealed that the manipulation likewise affected all three dimensions of trust.

At the same time, it is possible that our manipulation not only affected trust in science but also unintentionally influenced other constructs, which may have counteracted its intended effects (i.e., confounding). This seems particularly likely because our trust in science manipulation relied on factual statements from real scientists. While this increases the external validity and real-world relevance of our manipulation, it means that we could not perfectly parallelize the statements between conditions, as real statements naturally differ on various dimensions. Although we ensured that all statements covered similar topics, subtle differences between stimuli may have influenced other psychological variables, potentially overshadowing the effects of trust in science[21]. However, we tested this concern for a range of measured variables (e.g., fear or mood) and found no evidence that our manipulation systematically impacted them, except for trust in science. Nevertheless, our manipulation could have affected other constructs that we did not investigate. Future research could address this concern by using different manipulations, for example, by providing only the abstract rate of correct predictions[23] instead of specific statements.

Beyond this, it should be noted that other conceptualizations of trust in science (beyond trust in scientists) exist. For example, trust in science has been described as epistemic trust, focusing more strongly on scientific knowledge[15], or as institutional trust, referring to the institutionalized nature of science[38,53]. Finally, trust in science can also be understood as trust in the scientific method and process itself[42,53]. These different perspectives suggest that our manipulation may have been too narrow to fully alter such a heterogeneous construct as trust in science, which may, however, be reflected in the correlational results observed earlier. Future research may therefore need to manipulate various aspects of trust in science to test their causal effects more broadly.

Another possibility is that causal effects of trust in science do exist, but they are absent in the specific national context we studied. We only tested our manipulation in one country (i.e., Germany); thus, it is unclear whether our results can be generalized to other nations. This may be particularly relevant in the context of the COVID-19 pandemic, where national politics may strongly influence trust and protection intentions. Nevertheless, it seems unlikely that the correlation between trust in science and protection intentions, observed consistently across countries[6], would reflect a causal effect everywhere except in Germany. In fact, Germany seems to be an average country in terms of trust in science. Studies involving multiple countries found that Germany ranked mid-range in general trust in science[54] and in the link between trust in science and protection intentions[6]. Nevertheless, future research would benefit from examining diverse national contexts. Moreover, such research could use fully representative samples, going beyond the quota samples (representative only for gender and/or age) employed in our studies.

Such truly representative samples could also be valuable for studying heterogeneity in audience effects. It seems possible that our manipulation only influenced trust in science among certain groups, which could limit its effectiveness. For example, in light of political polarization, the manipulation may be less effective for individuals who perceive themselves as ideologically dissimilar to scientists[55], or for people who endorse conspiracy theories[56]. Future research could therefore employ pre–post designs with repeated measures of trust in science to more clearly identify which demographic or attitudinal groups are most responsive to such manipulations.

A final, but no less plausible, possibility is that the effects of trust in science exist at a relatively stable trait level, but we only manipulated state-level trust, which fluctuates with situational factors[25]. It is possible that typical trust-in-science research using questionnaires and correlational designs primarily captures the trait component. In contrast, manipulating trust in science may primarily affect the short-term situational component of trust in science without impacting the underlying trait. Perhaps such short-lived manipulations cannot alter protection intentions, which are likely formed over a more extended period and are thus more strongly influenced by trait levels of trust. However, such more enduring effects are unlikely to be captured by typical brief experimental studies and may require different methods, such as longitudinal studies[57], network models[58], or causal modelling[19]. Such approaches could test alternative ideas beyond the direct causal effect of trust in science, for example, by treating trust in science as a mediator or moderator, as done in some previous work[59–61].

Yet, classic mediation and moderation analyses have many pitfalls related to underlying causal inference problems[62]. Future research may thus benefit from engaging more explicitly with causal inference frameworks, particularly directed acyclic graphs (DAGs)[19,63]. DAGs are a graphical notation that facilitates systematic reasoning about complex causal structures. They can clarify, for example, which covariates should be adjusted for, which can safely be ignored, and in which cases statistical control may bias rather than improve causal inference. When combined with appropriate domain knowledge to ensure that all relevant variables are assessed, DAGs thus offer a powerful tool for improving the validity of causal claims based on correlational data[19]. Future cross-sectional research should therefore aim to carefully include relevant variables to construct appropriate DAGs that may accurately reflect the underlying causal processes.

From an applied perspective, this explanation would further suggest that short-term interventions to increase trust in science are ineffective in tackling pandemics. Instead, fostering long-term trust in science as a trait would be essential, which may require strategies such as improving education, enhancing science literacy, or increasing exposure to scientific principles[4]. Moreover, such efforts could also foster a broader understanding of the scientific process and its inherent uncertainty, which may help mitigate the adverse effects of incorrect predictions on trust in science[42]. Testing the effectiveness of such long-term interventions would require significant effort. Still, it could be a promising avenue for future research on the effects of trust in science, including its impact on protection intentions and beyond.

## Conclusion

Albeit speculative, it thus seems possible that long-term interventions targeting trust in science could positively affect protection intentions.

Nevertheless, our findings cast doubt on the immediate importance of trust in science for protection intentions. If trust in science does not immediately influence protection intentions, altering trust in science should not be the primary focus of urgent interventions in future pandemics. Our evidence suggests that it is unlikely that short-term changes in trust in science would alter intentions, let alone behavior. Perhaps other sources of information, such as politicians[4,64,65], family, and friends[66], or in the future, even artificial intelligence advisors[67], are more important levers to target in future pandemic behavior. For example, past work found that the correlation between trust in science and protection intentions is reduced when accounting for social conformity[68], suggesting that other social information might also play an important role. Likewise, it also seems possible that interactions need to be considered more broadly. For example, trust in science may interact with other variables, such as fear of infection or trust in politics, in shaping protection intentions and behaviors[60]. These ideas, however, again need to be tested using experimental methods.

On a more general note, our findings once again call for more caution when using correlational designs to detect causal effects, as this may lead to wrong conclusions. Even in the context of a pandemic, where rapid research is needed, and correlational designs may be the most feasible, these designs should be at least followed up by experimental approaches as soon as possible. Systematic analyses revealed that COVID-19 research, in general, was more likely to be correlational than other research[2,69], suggesting that many other findings likewise need experimental validation. This experimental work remains vital to draw the correct lessons from the pandemic.

On a positive note, our findings could reassure scientists engaging in public science communication during pandemics. At the beginning of a pandemic, scientific knowledge will always be characterized by strong uncertainty, and it is almost unavoidable that some predictions turn out wrong[42]. While we observed that wrong statements of scientists at the early stage of a pandemic can reduce trust in science (i.e., our trust in science manipulation), we found no evidence that this has actual consequences related to protection intentions. While it remains essential to be mindful of uncertainty in science[70], our results suggest that concerns about adverse consequences of premature science communication may be unfounded.

Taken together, it is always difficult to ultimately prove the absence of an effect[35,51]. Nevertheless, our results cast doubts on whether trust in science is a relevant causal driver of protection intentions during a pandemic. Based on our findings, it seems unlikely that short-term changes in trust in science will lead to meaningful changes in protection intentions. Although building trust in science likely has many positive effects, an immediate, short-term impact on pandemic behavior may not be one of them. Instead, in light of our findings, future policymakers and researchers should focus on other tools to curb the spread of pandemics. Identifying these tools and lessons from the COVID-19 pandemic, with careful attention to causal relations, remains essential for future research and policy.

## Data availability
Our data are available on the Open Science Framework (OSF) at https://osf.io/we6pk/.

## Code availability
The R code used for data analysis is available at https://osf.io/we6pk/.

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

## Acknowledgements
This research was supported by a DFG Grant (DO 1900/4-1) awarded to Simone Dohle. The funders had no role in study design, data collection and analysis, decision to publish or preparation of the manuscript. We thank Antonia Dörnemann, Cheyenne Eßer, and Britta Weis for their valuable support with the manuscript.

## Author contributions
S. Dohle generated the idea for the research project, with feedback from A.C. Posten and T. Wingen. S. Dohle programmed the study and collected the data, with feedback from A.C. Posten and T. Wingen. S. Dohle and T. Wingen wrote the analysis code and analyzed the data. T. Wingen wrote the first draft of the manuscript, and all authors critically edited it. All of the authors approved the submission of the final manuscript.

## Funding

## Competing interests
The authors declare no competing interests.
