## [Transparent Peer Review file · Communications Psychology]

No Evidence for Causal Effects of Trust in Science on Intentions for Health-Related Behavior

Corresponding Author: Professor Simone Dohle

Version 0:

Decision Letter:

Dear Professor Dohle,

Thank you for your patience during the peer-review process. Your manuscript titled "No Evidence for Causal Effects of Trust in Science on Pandemic-Related Protection Intentions" has now been seen by 2 reviewers, and I include their comments at the end of this message.

One reviewer vetted the original version of the manuscript and the present revision, the other reviewer was not involved in the first assessment. The reviewer from the original panel finds your work suitably improved, the other referee mentions some concerns that will need to be addressed.

In addition to the referees' concerns, there are also editorial points that must be resolved in a comprehensive revision.

In sum, we are interested in the possibility of publishing your study in Communications Psychology, but would like to consider your responses to these concerns and assess a revised manuscript before we make a final decision on publication.

We therefore invite you to revise and resubmit your manuscript, along with a point-by-point response to the reviewers. Please highlight all changes in the manuscript text file.

The editorial concerns revolve mostly around the reporting of the statistical analyses, in particular support for the null findings.

For manuscripts that report null results, we require:

- Evidence that the study was sufficiently powered to detect the smallest theoretically or pragmatically meaningful effect
- Bayes Factors or equivalence tests to interpret the null results that are reported in the main paper.

We therefore ask that in revision, you include both sensitivity power analysis and equivalence tests in the manuscript, rather than the SI.

We also ask that you ensure the formatting of the paper is fully compliant with the Communications Psychology Article format, as listed in the attached Editorial Requests Table. This checklist details critical reporting requirements for the revised manuscript. For example, you must reorder the manuscript to include the Methods before you report the Results.

Please attend to each item and ensure your manuscript is fully compliant. If your revised manuscript is not aligned with these requests on major issues, such as those concerning statistics, it may be returned to you for further revisions without re-review.

Please submit the following items:

- Revised manuscript

- Point-by-point response to the referees' comments
- Cover letter (as a separate document)
- [Nature Research Reporting Summary](https://www.nature.com/documents/nr-reporting-summary.pdf)
- Completed Editorial Request Table (attached).

via this link: Link Redacted .

Additional guidance is available in our style and formatting guide [Communications Psychology formatting guide](https://www.nature.com/documents/commpsychol-style-formatting-guide-accept.pdf).

Best regards,

Marika Schiffer

Marika Schiffer, PhD
Chief Editor
Communications Psychology

REVIEWER REPORTS:

Reviewer #1 (Remarks to the Author):

I am satisfied with the authors' responses to my previous review.

Reviewer #2 (Remarks to the Author):

Before getting into the review, just a quick note: if I only get invited to review a paper after it's already been through a round of reviewing (as seems to be the case here, since the submission includes a response to reviewers), I think it only fair to the authors if I constrain my comments to more minor issues, rather than suggesting major departures for a manuscript that has already been through substantial changes. So I'm purposefully not requesting major re-analyses, even if I think they might be useful.

I think this paper is a useful illustration of how the "trust in science" literature needs more nuance and refinement after the pandemic drew so much attention to the role of trust. I would like to eventually see it published and I think it is suitable for this journal.

My main concern is about how the core manipulation is presented. It is often framed as a manipulation of trust in science in some general sense (p11: "this manipulation affects trust in science"; p20: "Despite successfully altering people's trust in science"). I realise that the authors sometimes present more nuanced framings and caveats, but I think the main message still swings too far towards this overly general framing. When discussing this specific manipulation and its effects, I'd recommend something more concrete and limited than the above quotations.

In particular, this could include more prominent mention of "rated" or "self-reported" or "momentary" trust in science. As the general discussion already contrasts trait and state ("momentary") views of the broader phenomenon, perhaps the first two of these options might be more helpful as additions when describing this manipulation.

Other potential ways forward would be to highlight that the manipulation concerns "trust in Covid claims" or something similar. The latter point is intended to highlight how this "trust in science" manipulation is about the eventual fate of claims produced by Covid scientists, as contrasted with the process of doing science (and such claims are just one output of such a process). By defocusing the process aspect of science (including, for instance, how science works well when it is self-correcting, so a change in consensus truth status of claims---a key aspect of the manipulation---really just reflects science working as it should), the manipulation misses a crucial part of trust in science, and this could be better reflected in the framing of the manipulation.

The above point may also be relevant for the general discussion (e.g., p23 "fostering long-term trust in science"), where long-term trust may need more focus on process rather than isolated claims.

My other main comment---boosting a request for more theory that another reviewer made in the previous round---is to step back and consider what trust in science is meant to be or achieve. Specifically, where p21 includes some potential explanations ("we failed to observe it because it is too small, our measures are too noisy, or our sample size is insufficient"), these are mainly focused on methodological, psychometric or analytic issues, rather than on issues to do with the very concept being examined: trust in science.

Part of the problem here could be that trust in science is too ill-defined (in the literature generally, not necessarily meant as a criticism of this manuscript) or too heterogeneous to be amenable to this kind of experimental manipulation. Relatedly, there may be heterogeneity in audience effects (only some people might actually and sincerely be persuaded by your manipulation to really change their trust in science in a meaningful rather than momentary or superficial sense).

As I mentioned, I won't ask for major new directions in a second round of review, so I'm only suggesting the following as limitations/future directions for the general discussion, rather than things that need to be handled in the next revision. Both suggestions concern best practice in causal inference.

(1) Related to the mention of audience heterogeneity above, it would be useful to measure trust in science before and after the manipulation. If only certain people are impacted by the manipulation (while others remain resolutely pro- or anti-science) then this would put an upper limit on how large of an effect we might expect.

(2) The authors recognise that trust in science and relevant covariates present a complex picture. But this is precisely why the causal inference literature recommends Directed Acyclic Graphs (DAGs) as a way to make your assumptions about the causal structure explicit, also using these to guide your choice of modelling. The authors cite Julia Rohrer at some point, but DAGs are a core part of her message and these are sidestepped here.

As a key message of this paper is about issues around causation, I really think it needs to engage better with the causal inference literature, even if this is just in the general discussion.

* TRANSPARENT PEER REVIEW: Communications Psychology uses a transparent peer review system. This means that we publish the editorial decision letters including Reviewers' comments to the authors and the author rebuttal letters online as a supplementary peer review file. However, on author request, confidential information and data can be removed from the published reviewer reports and rebuttal letters prior to publication. If your manuscript has been previously reviewed at another journal, those Reviewers' comments would not form part of the published peer review file.

Version 1:

Decision Letter:

Dear Professor Dohle,

Your manuscript titled "No Evidence for Causal Effects of Trust in Science on Pandemic-Related Protection Intentions" has now been seen by our reviewers, whose comments appear below. In light of their advice I am delighted to say that we are happy, in principle, to publish a suitably revised version in Communications Psychology.

We therefore invite you to revise your paper one last time to address the remaining concerns of our reviewers and a list of editorial requests. At the same time we ask that you edit your manuscript to comply with our format requirements and to maximise the accessibility and therefore the impact of your work.

EDITORIAL REQUESTS:

SUBMISSION INFORMATION:

OPEN ACCESS:

*** TRANSPARENT PEER REVIEW:** Communications Psychology uses a transparent peer review system. On author request, confidential information and data can be removed from the published reviewer reports and rebuttal letters prior to publication. If you are concerned about the release of confidential data, please let us know specifically what information you would like to have removed. Please note that we cannot incorporate redactions for any other reasons.

*** CODE AVAILABILITY:** All Communications Psychology manuscripts must include a section titled "Code Availability" at the end of the methods section. We require that the custom analysis code supporting your conclusions is made available in a publicly accessible repository at this stage; please choose a repository that generates a digital object identifier (DOI) for the code; the link to the repository and the DOI must be included in the Code Availability statement. Publication as Supplementary Information will not suffice.

*** DATA AVAILABILITY:**

Link Redacted

Best regards,

Marika

Marika Schiffer, PhD
Chief Editor
Communications Psychology

REVIEWERS' COMMENTS:

Reviewer #2 (Remarks to the Author):

I thank the authors for their constructive engagement with the previous round of feedback. I think the manuscript is much improved, and think it is suitable for publication pending some minor issues that remain.

In particular, I appreciate that the authors added nuance regarding trust and its role. However, there are still a few places where the offered claims are more general than is warranted. E.g. p 20 "Study 2 found no evidence that the positive correlation between trust in science and protection intentions reflects a causal link." Again, just adding "self-reported" here would help highlight that participants' responses to this survey need not reflect the more state-like sense of trust that you later mention.

Similarly, on p29: "Another explanation for the absence of effects could be that while our manipulation does change trust in science in general, it may not have changed all relevant dimensions of this trust." Again, I don't think that this manipulation speaks to "trust in science in general", given the trait/state distinction (if both were affected, I'd be willing to see this as "general", but there's no evidence for that). Some mild weakening of the phrasing would be enough to render this unobjectionable.

The remaining comments are mainly aimed at improving flow and clarity.

The end of the introduction outlines some key motivations for each study (e.g., fear, fictional future pandemic), but these are not very prominent when each study is later described. E.g., on p12 for study 3, there's no reminder about fear except for a brief mention on p13 that this is a new variable. Which isn't enough to highlight for the reader that this is the whole point of study 3. Similarly, there's no mention of fictional future pandemics on p14 when study 4 is described. Provide a clear reminder for why you ran each study when its design/procedure is explained.

Figure 1 might benefit from some jittering (or some more transparency in the points) to better show where numerous data points overlap. If there are actually few overlaps, you can ignore this.

On p31, you mention a "final possibility". This seems the most compelling possibility to me! But you needn't agree with my assessment. Still, a minor tweak (e.g., "a final, but no less plausible, possibility") could make it clear that this is "last but not least".

Point by Point Response to Reviewers' Comments

Reviewer #1 (Remarks to the Author):

Reviewer 1 Comment 1. It's an interesting paper and I applaud the authors for having conducted such a timely experimental study on covid – this was a unique opportunity at the early stages of the pandemic. I also particularly like the clarity of the research, the fact that all materials and code is openly shared.

Response Reviewer 1 Comment 1:

We thank the reviewer for the positive and encouraging feedback. We sincerely appreciate the kind words regarding the timeliness of the study and the clarity of our research. We are also grateful that the reviewer acknowledged our commitment to transparency by sharing all materials and code openly. We finally appreciate all the helpful comments, which have strengthened the manuscript.

Reviewer 1 Comment 2. The data are interesting and in my view hold more potential than is currently used. I understand and appreciate that the authors followed a preregistered plan, but I think they could have explored the patterns in the data more than currently done.

*Based on the shared data, I did a quick mediation analysis on condition  trust in science  and the protective intention outcomes, based on the combined data from study 2-4. There I clearly found evidence for the indirect effects, yet also a substantial *negative* direct effect of the manipulation on PI across all four outcomes (see screenshot below). It seems these cancel*

each other out, resulting in non-significant total effects across all PI measures. (This also becomes evident in (multilevel) linear regressions on the aggregated data, where for all four DVs both trust and condition, as well as the interaction between them, are significant predictors. This indicates that keeping trust in scientists constant, there is a negative effect of the manipulation on PI).

*Having read the trust manipulation items, I wonder if there is a confound that may explain this pattern of results, namely that showing incorrect information, and pointing out that it is incorrect, may actually enhance belief in the need for/importance of action. This is perhaps somewhat similar to the idea of psychological inoculation (see the work of Sander van der Linden) as a means to convince people of scientific facts (as opposed to believing misinformation). In this specific case, I believe the false statements were mostly downplaying the virus, saying that it isn't that dangerous and that it will be over soon. By pointing out that these positive prospects were in fact false, it may have only made people more aware of the need for protection and warned them to not underestimate the virus. Having been shown that a lot of incorrect information was initially communicated by scientists may cause people to report that apparently, German scientists weren't always right about the virus (hence lower trust scores). Yet in the end, these messages underestimating the risks of the virus were discovered to be incorrect (as already shown in the manipulation), which actually boosted people's awareness of the need for protective actions (because it wasn't a small proportion that got sick and it is more dangerous than influenza). I realize that the authors investigated this possibility by looking at the mediating effects of fear, perceived danger and severity of the virus in Study 3, and did not find an effect there. Yet, the fact that a mediation analysis on all data indicates a significant positive indirect effect via trust *and* a significant negative direct effect (which cancel each other out in the total effect) cannot be ignored I believe. Perhaps it's not so much about people being more afraid that they will personally suffer from Covid (as the mediation items focus on), but*

that, contrary to early messages, the scientific consensus now exists that Covid can be very harmful to vulnerable groups.

Essentially, these data show that the manipulation does change trust in science in the expected direction (XM) and that trust in science is strongly correlated with protective intentions for Covid (MY). However, there is no direct effect of the manipulation on PI. So whatever form of trust in science is manipulated by the statements, it is not the part that also subsequently changes PI. The authors' explanation of the difference between situational and trait-level evaluation of science might be important here. Could the difference between scientists as fallible humans and science as a credible method/system also be relevant here? So people acknowledge that scientists can also be wrong (hence lower trust scores), yet they do still believe in the ability of science to arrive at "true" facts because they trust and have faith in the scientific system as a whole. Both conditions of the manipulation demonstrate the self-correcting nature of science in the sense that incorrect statements were ultimately corrected (by other scientists). In other words, perhaps the manipulation did not effectively undermine the credibility of science.

Response Reviewer 1 Comment 2:

We thank the reviewer for this detailed and thoughtful analysis. We appreciate the deep engagement with the data.

Like the reviewer, we also deemed the possibility of confounding a critical issue. This is why we investigated a variety of possible confounds in Study 3 (i.e., fear, different facets of mood, pandemic fatigue, or perceived vulnerability or severity of COVID-19), but found no evidence for confounding (see Study 3). As the reviewer notes, if our manipulation made the virus appear more dangerous, we would expect effects on, for example, fear or severity of COVID-19, but observed none. That said, we agree that other unmeasured mechanisms may be at play. We now discuss these possibilities more clearly in the revised manuscript (p. 20) and write:

[...] we tested this concern for a range of measured variables (e.g., fear or mood) and found no evidence that our manipulation systematically impacted them, except for trust in science. Nevertheless, our manipulation could have affected other constructs that we did not investigate. Future research could address this concern by using different manipulations, for example, by providing only the abstract rate of correct predictions²³ instead of specific statements.

Regarding the mediation pattern, it is often helpful to include the manipulation check as a mediator (the so-called *TMC* procedure suggested by Lench et al., 2014). However, it should be noted that these authors explicitly argue that it is not meaningful to conduct a *TMC* when the experimental condition has no effect on the dependent measure (Lench et al., 2014, p. 220).

In fact, the negative direct effect of our manipulation is likely a simple statistical artifact of including trust in science as a mediator. As our manipulation affects trust in science, and trust in science is generally correlated with protection intentions, this will necessarily lead to a positive indirect effect. As the total effect is zero, this in turn **must lead** to a negative direct effect, which, however, simply reflects that our manipulation affects the manipulation check (correlated with the dependent variable) but not protection intentions. If interested, we have attached R code demonstrating this phenomenon at the end of this point-by-point response letter.

***Reviewer 1 Comment 3.** In any case, if the authors truly believe that trust in science is not causally related to PI, could they reflect on what confounders may give rise to the rather strong observed correlation in the data. Would something like level of education, or epidemiological knowledge be a common cause of both trust in science and PI? Perhaps longitudinal studies and causal modeling may also prove informative in this domain. I believe it would be worthwhile for the authors to discuss these possibilities and complementary methods.*

Moreover, it might also be relevant to discuss the limits on the ‘holy grail’ of experimental research in the context of this study. Although brief experimental manipulations may be capable of shifting some aspects of perceptions and beliefs, they may not be powerful enough to effectively alter personally consequential (intentions of) behavior, such as health behavior. And maybe we should be worried if they were? That is, in a way, we might be happy that protective intentions, including vaccination intentions aren’t easily reduced by a 5-minute exposure to incorrect scientific information. I’d invite the authors to reflect on these ideas and perhaps suggest future avenues for addressing these types of causal questions.

Response Reviewer 1 Comment 3:

We now reflect on this point in the General Discussion (p. 21), noting the limits of brief experimental manipulations and highlighting the value of longitudinal and causal modeling:

However, such more enduring effects are unlikely to be captured by typical brief experimental studies and may require different methods, such as longitudinal studies⁴¹, network models⁴², or causal modelling¹⁹. Such approaches could test alternative ideas beyond the direct causal effect of trust in science, for example, by treating trust in science as a mediator or moderator, as done in some previous work⁴³⁻⁴⁵.

Regarding the reviewer's positive interpretation that a 5-minute intervention does not diminish protective intentions, we have added this point to the discussion (p. 22).

On a positive note, our findings could reassure scientists engaging in public science communication during pandemics. At the beginning of a pandemic, scientific knowledge will always be characterized by strong uncertainty, and it is almost unavoidable that some predictions turn out wrong⁵². While we observed that wrong statements of scientists at the early stage of a pandemic can reduce trust in science (i.e., our trust in science manipulation), we found no evidence that this has actual consequences

related to protection intentions. While it remains essential to be mindful of uncertainty in science⁵³, our results suggest that concerns about adverse consequences of premature science communication may be unfounded.

***Reviewer 1 Comment 4.** Finally, I believe the machine learning analysis is not optimal nor particularly relevant. While the idea of machine learning sounds nice, I wonder what “complex” interaction patterns the authors were expecting? Essentially, I think it’s better to critically assess the materials and try to understand what may drive these results from a causal perspective than seeing if a naïve machine learning algorithm may discover something (potentially uninterpretable). I am not a machine learning expert though, so of course I might be missing some important reason for applied it to these data. In that case, I’d like to see some more justification and explanation of what the machine learning approach may have discovered.*

Response Reviewer 1 Comment 4:

We included the machine learning analysis in response to earlier conference feedback suggesting that there might be complex effects not captured by standard models. Rather than hypothesizing post hoc (which could lead to problematic practices such as *HARKING*, Rubin, 2017), we used a random forest model to test for such patterns broadly. Random forests are well-suited for detecting interactions and non-linear relations. If we had observed any evidence for such patterns, we planned to follow up with a closer examination (e.g., by inspecting variable importance or by employing interpretable machine learning models, such as Lasso regression). We now clarify this rationale and approach more clearly (p. 30).

Finally, we considered the possibility that trust in science might not have a simple, direct causal effect on protection intentions. Instead, it seemed possible that the causal link

between trust in science and protection intentions is more complex and, for instance, moderated by other variables, such as age, gender, political orientation, or even more complex interaction patterns. In this case, trust in science would still have a causal, yet very complex, effect on protection intentions. Such complex relations, however, would be missed by many classic statistical methods, such as *t*-tests, but can be revealed by more advanced analyses. To avoid hypothesizing post hoc and selectively testing only specific moderations (e.g., in linear regressions), we instead employed a machine learning algorithm to explore potential interaction patterns more broadly.

Reviewer #1 (Remarks on code availability):

The code is clearly annotated, reproducible and understandable.

Reviewer #2 (Remarks to the Author):

*Reviewer 2 Comment 1. This is an interesting set of studies that tackles an important question. I very much agree with the authors that we should be careful to infer causality from correlational inquiries into the 'predictors' of protection intentions (note here that the explanation for this on p. 4 is a bit much - this is good for a textbook but I don't think the readership of this journal needs a correlation-causation 1.0 course). Things are probably more complex, with individual differences in beliefs and ideologies, social norms, etc. potentially playing a role. Or, perhaps trust in science could be a (or one of many) moderators of effects of such variables (e.g., social norms) on intentions. Other (correlational) work points to trust in science as a mediator of the relation between spirituality and (some) protection intentions (Zarżeczna, N., Večkalov, B., & Rutjens, B. T. (2023). Spirituality and intentions to engage in Covid-19 protective behaviours. *Social and Personality Psychology Compass*, 17(8), e12765)*

Response Reviewer 2 Comment 1:

We thank the reviewer for their careful reading and constructive feedback. We appreciate the positive evaluation of our work and the insightful suggestions.

As recommended, we have shortened the correlation-causation explanation in the introduction to ensure it is more concise and appropriate for the journal's audience.

Moreover, we like the idea that trust in science may play an important role as a mediator or moderator rather than as the causal driving force of protection intentions. While our machine learning approach would likely have detected some forms of moderation at least with regard to the variables included in the model, we still think that other models could be tested by future work. We added this idea on page 21:

[...] may require different methods, such as longitudinal studies⁴¹, network models⁴², or causal modelling¹⁹. Such approaches could test alternative ideas beyond the direct causal effect of trust in science, for example, by treating trust in science as a mediator or moderator, as done in some previous work⁴³⁻⁴⁵.

Reviewer 2 Comment 2.

The studies reported are well-conducted and convincing. Moreover, the General Discussion is very well done, offering various thoughtful reflections on the research. I agree with the authors' point (which is also in line with the machine learning results) that trust in science might be a particularly potent individual difference variable that is hard to experimentally move around on all levels.

However, what is missing from the manuscript and what I would think is needed to warrant publication in NHB is more than what the paper currently offers: WHAT IS IT then that helps causally predict protection intentions? For example, I wouldn't be surprised if trust in the government plays a role, perhaps interacting with trust in science (aforementioned moderation). Or perhaps it's more about people's trust in the ability of politicians to build trustworthy policies building on scientific evidence.

In short - while I don't think that the current results are enough to warrant publication in NHB, I do hope that the authors manage to publish this important work in a more fitting outlet.

Response Reviewer 2 Comment 2:

We thank the reviewer for their positive evaluation of our studies and the General Discussion.

We appreciate the acknowledgment of the relevance and quality of our work.

We fully agree that identifying causal predictors of protection intentions remains an important task. However, given the prominence of findings linking trust in science to protection intentions—including publications in high-impact journals such as *PNAS* and *Nature Human Behaviour* (e.g., Algan et al., 2021, *PNAS*; Sturgis et al., 2021, *NHB*)—we believe it is first crucial to publish evidence questioning the assumed causality of this relationship. Current policy recommendations often rely on this correlation.

In response to the suggestion, we have extended the discussion on potential causal factors beyond trust in science. We now write (p. 21 - 22):

[...] Perhaps other sources of information, such as politicians^{4,46,47}, family, and friends⁴⁸, or in the future, even artificial intelligence advisors⁴⁹, are more important levers to target in future pandemic behavior. For example, past work found that the correlation between trust in science and protection intentions is reduced when accounting for social conformity⁵⁰, suggesting that other social information might also play an important role. Likewise, it also seems possible that interactions need to be considered more broadly. For example, trust in science may interact with other variables, such as fear of infection or trust in politics, in shaping protection intentions and behaviors.⁴⁴ These ideas, however, again need to be tested using experimental methods.

Reviewer #2 (Remarks on code availability):

-

Reviewer #3 (Remarks to the Author):

Reviewer 3 Comment 1. I appreciate the opportunity to read this insightful and overall very well-done paper and provide my assessment of it, which I share below.

The authors present a systematic analysis of a claim that has often been tacitly assumed and probably affected many policy decisions and communication strategies, but has not been sufficiently tested; which is that trust in science was a predictor of protective measures during the COVID-19 pandemic. This alone is laudable, and so is the comprehensive analysis spanning over four studies, involving innovative, state-of-the-art methods, and, overall, much methodological rigor. Also, the paper is clearly written. However, I have two stronger concerns:

Response Reviewer 3 Comment 1:

We thank the reviewer for their thorough review and positive evaluation of our work, despite the concerns raised. We address the concerns below.

Reviewer 3 Comment 2. First, I believe that the studies have too many limitations that justify conclusions as strong and consequential as: “our results cast significant doubts on whether trust in science is a relevant causal driver of protection intentions during a pandemic”. The authors do use careful language throughout most of the manuscript, also, the title is appropriate. Moreover, they engage with many of these limitations thoroughly. Taken together, the most significant limitations include:

- As far as I see, the manipulations focused on manipulating perceived expertise of scientists, but not on perceived benevolence (acting in the best interest of society) or specifically on the perceived accuracy of evidence on the effectiveness of protective measures. Hypothesizing an effect of those general expertise-related treatments on protective behaviors seems to be too much of a stretch to me – even if the authors argue against this on p. 18. Why didn't the authors test other interventions?

- *The study samples barely represent the average population of Germany: They did not contain quotas for education (which likely has an effect on trust in science and possibly also on protective behaviors). In addition, the samples were recruited through online access panels, which might have had a slight effect on the results such as that participants might have been more likely than the average population to rely on online media to connect with others – and therefore, during the pandemic, less likely to engage with others and be in public spaces.*

- *Study 4 was done about 2 years into the pandemic, so participants likely had some pandemic fatigue that discouraged them to perform protective behaviors “in the next four weeks” (see questionnaire)*

- *There might have been ceiling effects for trust in science, as trust was relatively high in Germany and even increased in the first phase of the pandemic – so even if the manipulation check showed that trust increased due to manipulation, this increase might have been too marginal to effectively cause any behavioral changes*

- *The context was Germany, which had comparably lax containment measures and enforcement or legal sanctions for not following regulations (which might have caused many participants to just not care about protective measures)*

Response Reviewer 3 Comment 2:

We sincerely thank the reviewer for this detailed and thoughtful feedback. We appreciate the recognition of our efforts and the careful reflection on the limitations. We address each point below and have incorporated corresponding clarifications into the manuscript:

Too strong wording: We have removed/rephrased wordings where we argue that the results cast *strong or significant doubts* throughout the manuscript. In addition, we have removed the first part of the title, 'When Correlation Does Not Imply Causation,' which also seemed too strongly worded.

Manipulation focus: Our manipulation was selected based on a pre-study, where we found that it affects all three dimensions of trust in science—expertise, integrity, and, importantly, also benevolence. The effect on benevolence was smaller than on the other dimensions but still substantial ($d = 0.50$). We realize that this may have been hidden in the supplemental materials and have now added additional details on these results both to the supplemental materials and to the main manuscript, where we write (p. 10)

Details on the pretest, where we found that this manipulation affects trust in science both in general ($d = 0.74$) and in its different facets benevolence ($d = 0.50$), expertise ($d = 0.63$), and integrity ($d = 0.69$), can be found in the Supplemental Materials.

Sample representativeness: We agree that, despite quota sampling, our samples are not fully representative, particularly regarding education. We have now explicitly acknowledged this as a limitation (p. 20).

Nevertheless, future research would benefit from examining diverse national contexts. Moreover, such research could use fully representative samples, going beyond the quota samples (representative only for gender and/or age) employed in our studies.

Pandemic fatigue: The reviewer correctly points out that our studies were conducted at a later stage of the pandemic. Thus, it is essential to consider whether our manipulation may only be effective in situations of low pandemic fatigue. We hence reanalyzed our data from Study 3, where we measured pandemic fatigue, to find evidence for or against this claim.

First, we observed that pandemic fatigue was not extraordinarily high ($M = 4.8$, $SD = 1.7$, on a 7-point scale). Further, we observed no evidence that pandemic fatigue moderated the effect of our manipulation on any of the protection intentions, neither for social distancing ($p = .902$), hand hygiene ($p = .631$), mask-wearing ($p = .782$), or vaccination ($p = .699$).

We have added these analyses to the OSF report “Report Study 3”.

Ceiling effects: It is important to note that trust was measured on a 7-point scale. We have now moved this information to a more prominent position on page 24. Keeping this in mind, trust was not particularly high but rather close to the scale midpoint in our studies. We write:

Study 2:

Participants in the high trust condition reported higher trust in science ($M = 5.0$, $SD = 1.5$) than participants in the low trust condition ($M = 4.3$, $SD = 1.4$),

Study 3:

Participants in the high trust condition reported higher trust in science ($M = 4.5$, $SD = 1.6$) than participants in the low trust condition ($M = 3.8$, $SD = 1.5$)

Study 4:

Participants in the high trust condition reported higher trust in science ($M = 5.9$, $SD = 1.0$) than participants in the low trust condition ($M = 5.0$, $SD = 1.1$)

German context: We agree that contextual factors are relevant and now write (p. 20):

We only tested our manipulation in one country (i.e., Germany); thus, it is unclear whether our results can be generalized to other nations. This may be particularly relevant in the context of the COVID-19 pandemic, where national politics may strongly influence trust and protection intentions.

And later

Nevertheless, future research would benefit from examining diverse national contexts.

However, as noted in the revised discussion (p. 20), these factors would likely reduce both the correlation and the experimental effects. Given that Germany shows an average correlation between trust in science and protection intentions, we are not convinced that the national context alone could explain the absence of causal effects. Furthermore, one could even argue that the effects of trust in science should be particularly pronounced in countries with relatively low enforcement or sanctions (e.g., Germany), as strict sanctions might suppress such effects by compelling compliance regardless of individuals' trust levels.

Taken together, we hope these clarifications address your concerns and further strengthen the manuscript.

Reviewer 3 Comment 3 *Second, in my view, the study lacks theoretical rigor. While the authors do provide some worthwhile conceptual reasoning, e.g. on whether trust as state vs. trait, in the discussion section, I strongly encourage the authors to engage in such reasoning in the front-end of the paper.*

- In general, I would like to hear more theorizing why trust and protective behavior should be correlated in the first place and on why protective behavior might actually be a meaningful predictor of trust.

- Moreover, it is important to distinguish trust in science (as an institution, profession, repertoire of methods and theories, etc.) from trust in scientists (i.e. their perceived expertise, integrity, benevolence, etc.) and trust in the behaviors of scientists (do high-quality research, give policy recommendations, provide evidence for consumer choices etc, which is the approach underlying the operationalization the authors used).

*- More specifically, I think it is important to outline different approaches to conceptualize trust early on in the paper – and distinguish between those that see trust in science as a fundamental, relatively stable, long-term socialized value proposition (trait) and as a more variable context- and time-specific attitude to science (state). This distinction is important to contextualize the ability to *predict* attitudes, emotions, reasoning, and, specifically: protective behaviors. I'm aware that space is very limited, but I encourage the authors to engage with this literature in the front-end of the paper, not only in the discussion section.*

Response Reviewer 3 Comment 3:

We thank the reviewer for this thoughtful and constructive comment. However, we see multiple ways to frame this work. In its current form, the manuscript primarily focuses on re-evaluating the causal interpretation of a widely reported correlation between trust in science and protection intentions—an association frequently highlighted in high-impact publications and with clear relevance for policy-making. We aimed to address this applied question by systematically testing the assumed causal link. Therefore, we do not believe the manuscript should primarily focus on making a strong theoretical point. Nevertheless, we agree that the previous version still fell too short on theory, and we thus now have added two full paragraphs to the introduction to increase our theoretical rigor. First, we argue why trust in science may causally lead to protection intentions and behavior (p. 4):

From a theoretical perspective, such a causal interpretation could be seen as reasonable. Trust is typically defined as a willingness to make oneself vulnerable to another in anticipation of beneficial outcomes^{15,16}. In this sense, many recommended protective behaviors involve placing oneself in a vulnerable position by following scientists' advice (for example, by accepting a potentially harmful vaccine) with the expectation of a positive outcome, such as gaining immunity against infections. Therefore, individuals may be more likely to engage in scientifically recommended behaviors if their trust in science is high. This also aligns with classic models of persuasion, which emphasize that messages from trustworthy sources are more persuasive^{17,18}.

We further explain our conceptualization of trust in science with greater clarity on page 5:

It is, however, essential to recognize that trust in science can be conceptualized on multiple levels, such as trust in science as an institution, trust in scientists, or trust in scientific methods²³. Our studies focused specifically on trust in scientists, as they are likely the most visible aspect of science to the public. Another important distinction is between trait (i.e., trust in science as a relatively stable personality disposition) and

state-level components of trust (i.e., trust fluctuating with situational factors). Recent work suggests that, like most psychological constructs²⁴, trust encompasses state and trait components²⁵. Given the brief and experimental nature of our intervention, we likely primarily influenced the situational, state-level trust in science. We return to this distinction in the discussion section.

Reviewer 3 Comment 4. Minor further comments:

- To better embed the paper in the pertinent science communication/attitudes literature, I suggest referencing the debate over whether trust in science (or acceptance, sympathy, etc.) is a predictor or outcome of public understanding about science (or knowledge, awareness etc.; see Roberts et al., 2013).

- It would be helpful to also engage with literature on the link between trust and behavior beyond the COVID-19 pandemic, e.g., on the impact of trust in (climate) science and pro-environmental behavior (see Haltinner & Sarathchandra, 2022)

- Given that the authors challenge the causal link of trust and protective behaviors, I suggest to tone down the causal language in the literature review from “Later work confirmed that trust in science predicted individuals’ protection intentions...” to “Later work suggested that trust in science predicted individuals’ protection intentions...”

- This would need a reference: “At the same time, damaging trust in science (e.g., by raising unjustified doubts about science and scientists) could harm protection intentions.”

Response to Minor Comments Reviewer 3:

We thank the reviewer for these helpful suggestions to improve the positioning and clarity of our manuscript. In response, we have revised the literature review to tone down causal language

where appropriate. Further, we have rephrased the statement on “damaging trust in science” to clarify that this follows from an assumed causal effect.

Regarding the suggestion to further embed the introduction in the broader science communication and attitudes literature (e.g., Roberts et al., 2013; Haltinner & Sarathchandra, 2022), we acknowledge the relevance of these perspectives. However, as the revision is now done for a different journal (*Nature Communications*), we would prefer to await editorial guidance before substantially altering the theoretical scope of the introduction with regard to the minor comments.

References by Reviewer 3

Haltinner, K., & Sarathchandra, D. (2022). Predictors of Pro-environmental Beliefs, Behaviors, and Policy Support among Climate Change Skeptics. Social Currents, 9(2), 180–202. <https://doi.org/10.1177/23294965211001403>

Roberts, M. R., Reid, G., Schroeder, M., & Norris, S. P. (2013). Causal or spurious? The relationship of knowledge and attitudes to trust in science and technology. Public Understanding of Science, 22(5), 624–641. <https://doi.org/10.1177/0963662511420511>

Reviewer #3 (Remarks on code availability):

I did not spot errors in the code and was able to run it. However, I have to mention that my experience with the TOSTER package and internal meta-analysis is limited, and I have no expertise on the machine learning approach.

I commend the authors on well-written code and a clearly structured repository containing all necessary materials to replicate results.

References Point-by-Point Response

- Lench, H. C., Taylor, A. B., & Bench, S. W. (2014). An alternative approach to analysis of mental states in experimental social cognition research. *Behavior Research Methods*, *46*, 215-228. <https://doi.org/10.3758/s13428-013-0351-0>
- Algan, Y., Cohen, D., Davoine, E., Foucault, M., & Stantcheva, S. (2021). Trust in scientists in times of pandemic: Panel evidence from 12 countries. *Proceedings of the National Academy of Sciences*, *118*(40). <https://doi.org/10/gmxpcd>
- Rubin, M. (2017). When Does HARKing Hurt? Identifying When Different Types of Undisclosed Post Hoc Hypothesizing Harm Scientific Progress. *Review of General Psychology*, *21*(4), 308–320. <https://doi.org/10.1037/gpr0000128>
- Sturgis, P., Brunton-Smith, I., & Jackson, J. (2021). Trust in science, social consensus and vaccine confidence. *Nature Human Behaviour*, 1–7. <https://doi.org/10.1038/s41562-021-01115-7>

Appendix

Example R Code related to Response Reviewer 1 Comment 2:

```
# SIMULATION CODE
```

```
set.seed(123) # For reproducibility
```

```
n <- 500 # Number of observations
```

```
#simulate an unaltered mediator (mediator before experimental intervention, e.g. participants  
default trust in science)
```

```
unaltered_mediator <- rnorm(n)
```

```
# Simulate dependent variable (protection intention) predicted by unaltered mediator but NOT  
directly by condition (reflecting correlation but not causation)
```

```
dv <- 10 + 1.2*unaltered_mediator + rnorm(n)
```

```
#show significant correlation
```

```
cor.test(dv, unaltered_mediator)
```

```
# Simulate condition (binary: 0 = control, 1 = treatment)
```

```
condition <- rbinom(n, 1, 0.5)
```

```
# Simulate mediator affected by condition (i.e., trust in science is shaped by manipulation, but NOT protection intentions)
```

```
mediator <- 0.8*condition + rnorm(n, 0, 1) + unaltered_mediator
```

```
#show significant correlation
```

```
cor.test(dv, mediator)
```

```
#show null effect of condition on dv
```

```
t.test(dv~condition)
```

```
# Combine into data frame
```

```
sim_data <- data.frame(condition = condition,
```

```
    mediator = mediator,
```

```
    dv = dv)
```

```
# LAVAAN MEDIATION ANALYSIS
```

```
library(lavaan)
```

```
# Specify mediation model
```

```
mediation_model <- '  
  
# Direct effect (condition -> dv)  
  
dv ~ c*condition  
  
# Mediator path (condition -> mediator)  
  
mediator ~ a*condition  
  
# DV path (mediator -> dv)  
  
dv ~ b*mediator  
  
# Indirect effect (a*b)  
  
indirect := a*b  
  
# Total effect (direct + indirect)  
  
total := c + (a*b)  
,  
  
# Fit the model (not using bootstrapping to speed this up)  
  
fit <- sem(mediation_model, data = sim_data)
```

```
# View results with standardized estimates
```

```
summary(fit, standardized = TRUE)
```

```
#note that we observe exactly the pattern as in the manuscript: a negative direct effect, simply  
as a result of simulating an effect on the mediator but not on the DV
```

Point by Point Response to Reviewers' Comments

Reviewer #1 (Remarks to the Author):

Reviewer 1 Comment 1. I am satisfied with the authors' responses to my previous review.

Response Reviewer 1 Comment 1:

We sincerely thank the reviewer for their previous feedback and are grateful that our responses have been satisfactory.

Reviewer #2 (Remarks to the Author):

Reviewer 2 Comment 1. Before getting into the review, just a quick note: if I only get invited to review a paper after it's already been through a round of reviewing (as seems to be the case here, since the submission includes a response to reviewers), I think it only fair to the authors if I constrain my comments to more minor issues, rather than suggesting major departures for a manuscript that has already been through substantial changes. So I'm purposefully not requesting major re-analyses, even if I think they might be useful.

I think this paper is a useful illustration of how the "trust in science" literature needs more nuance and refinement after the pandemic drew so much attention to the role of trust. I would like to eventually see it published and I think it is suitable for this journal.

Response Reviewer 2 Comment 1:

We are very grateful that the reviewer found the paper useful and suitable for publication in this journal. We sincerely thank the reviewer for the constructive comments that have further strengthened the manuscript. We particularly appreciate the reviewer's considerate approach of not requesting major revisions for a manuscript that had already undergone substantial revision before this review.

Reviewer 2 Comment 2. *My main concern is about how the core manipulation is presented. It is often framed as a manipulation of trust in science in some general sense (p11: "this manipulation affects trust in science"; p20: "Despite successfully altering people's trust in science"). I realise that the authors sometimes present more nuanced framings and caveats, but I think the main message still swings too far towards this overly general framing. When discussing this specific manipulation and its effects, I'd recommend something more concrete and limited than the above quotations.*

In particular, this could include more prominent mention of "rated" or "self-reported" or "momentary" trust in science. As the general discussion already contrasts trait and state ("momentary") views of the broader phenomenon, perhaps the first two of these options might be more helpful as additions when describing this manipulation.

Other potential ways forward would be to highlight that the manipulation concerns "trust in Covid claims" or something similar. The latter point is intended to highlight how this "trust in science" manipulation is about the eventual fate of claims produced by Covid scientists, as contrasted with the process of doing science (and such claims are just one output of such a process). By defocusing the process aspect of science (including, for instance, how science works well when it is self-correcting, so a change in consensus truth status of claims---a key aspect of the manipulation---really just reflects science working as it should), the manipulation misses a crucial part of trust in science, and this could be better reflected in the framing of the manipulation.

The above point may also be relevant for the general discussion (e.g., p23 "fostering long-term trust in science"), where long-term trust may need more focus on process rather than isolated claims.

Response Reviewer 2 Comment 2:

In light of this important comment, we have revised the manuscript in several ways:

- a) We now consistently describe the outcome as self-reported trust in science where appropriate (e.g., when introducing the manipulation, when reporting its effects in the results, and in central positions such as the abstract and the general discussion).
- b) When introducing the manipulation, we explicitly highlight that it targeted one specific aspect of science, namely the accuracy of predictions about the pandemic made by scientists, rather than science more broadly or other aspects such as the scientific process. We write (p. 10):

Nevertheless, it is important to note that the manipulation solely focused on predictions made by COVID-19 scientists. In contrast, it did not address trust in science more broadly, nor aspects such as scientific methods or the scientific process, which inherently involve uncertainty and knowledge updating⁴².

c) In the general discussion, we added a new passage emphasizing that fostering long-term trust in science may also require building a broader understanding of the scientific process, including its inherent uncertainty and self-correcting nature. We write (p. 30):

[...] Instead, fostering long-term trust in science as a trait would be essential, which may require strategies such as improving education, enhancing science literacy, or increasing exposure to scientific principles⁴. Moreover, such efforts could also foster a broader understanding of the scientific process and its inherent uncertainty, which may help mitigate the adverse effects of incorrect predictions on trust in science⁴². Testing the effectiveness of such long-term interventions would require significant effort. Still, it could be a promising avenue for future research on the effects of trust in science, including its impact on protection intentions and beyond.

Reviewer 2 Comment 3 *My other main comment---boosting a request for more theory that another reviewer made in the previous round---is to step back and consider what trust in science is meant to be or achieve. Specifically, where p21 includes some potential explanations ("we failed to observe it because it is too small, our measures are too noisy, or our sample size is insufficient"), these are mainly focused on methodological, psychometric or analytic issues, rather than on issues to do with the very concept being examined: trust in science.*

Part of the problem here could be that trust in science is too ill-defined (in the literature generally, not necessarily meant as a criticism of this manuscript) or too heterogeneous to be amenable to this kind of experimental manipulation. Relatedly, there may be heterogeneity in audience effects (only some people might actually and sincerely be persuaded by your manipulation to really change their trust in science in a meaningful rather than momentary or superficial sense).

Response Reviewer 2 Comment 3:

We thank the reviewer for this comment, encouraging us to strengthen our manuscript's theoretical framing of trust in science. In response, we have expanded the discussion of what trust in science is meant to be, moving beyond methodological explanations for the absence of effects. Specifically, we now highlight different conceptualizations of trust in science (e.g., epistemic trust in scientific knowledge, institutional trust in the scientific system, and trust in scientific methods and processes). We now write (p. 28):

Beyond this, it should be noted that other conceptualizations of trust in science (beyond trust in scientists) exist. For example, trust in science has been described as epistemic trust, focusing more strongly on scientific knowledge¹⁵, or as institutional trust, referring to the institutionalized nature of science^{38,53}. Finally, trust in science can also be understood as trust in the scientific method and process itself^{42,53}. These different perspectives suggest that our manipulation may have been too narrow to fully alter such a heterogeneous construct as trust in science, which may, however, be reflected

in the correlational results observed earlier. Future research may therefore need to manipulate various aspects of trust in science to test causal effects more broadly.

Reviewer 2 Comment 4. *As I mentioned, I won't ask for major new directions in a second round of review, so I'm only suggesting the following as limitations/future directions for the general discussion, rather than things that need to be handled in the next revision. Both suggestions concern best practice in causal inference.*

(1) Related to the mention of audience heterogeneity above, it would be useful to measure trust in science before and after the manipulation. If only certain people are impacted by the manipulation (while others remain resolutely pro- or anti-science) then this would put an upper limit on how large of an effect we might expect.

(2) The authors recognise that trust in science and relevant covariates present a complex picture. But this is precisely why the causal inference literature recommends Directed Acyclic Graphs (DAGs) as a way to make your assumptions about the causal structure explicit, also using these to guide your choice of modelling. The authors cite Julia Rohrer at some point, but DAGs are a core part of her message and these are sidestepped here.

As a key message of this paper is about issues around causation, I really think it needs to engage better with the causal inference literature, even if this is just in the general discussion.

Response Reviewer 2 Comment 4:

We are again grateful for these thoughtful suggestions, which we have now integrated into the general discussion as future directions.

a) In line with the reviewer's point about audience heterogeneity, we expanded our discussion to note that our manipulation may have differential effects across subgroups. We further suggest that future studies could employ pre–post designs with repeated measures of trust in science. We write (p. 29):

Such truly representative samples could also be valuable for studying heterogeneity in audience effects. It seems possible that our manipulation only influenced trust in science among certain groups, which could limit its effectiveness. For example, in light of political polarization, the manipulation may be less effective for individuals who perceive themselves as ideologically dissimilar to scientists⁵⁵, or for people who endorse conspiracy theories⁵⁶. Future research could therefore employ pre–post designs with repeated measures of trust in science to more clearly identify which demographic or attitudinal groups are most responsive to such manipulations.

b) We also revised the discussion to engage more explicitly with the causal inference literature. In particular, we emphasize that traditional mediation and moderation analyses have important limitations, and we highlight the benefits of directed acyclic graphs as a framework for making causal assumptions transparent, including helpful references to Julia Rohrer's work. We write (p. 30):

Yet, classic mediation and moderation analyses have many pitfalls related to underlying causal inference problems⁶². Future research may thus benefit from engaging more explicitly with causal inference frameworks, particularly directed acyclic graphs (DAGs)^{19,63}. DAGs are a graphical notation that facilitates systematic reasoning about complex causal structures. They can clarify, for example, which covariates should be adjusted for, which can safely be ignored, and in which cases statistical control may bias rather than improve causal inference. When combined with appropriate domain knowledge to ensure that all relevant variables are assessed, DAGs thus offer a powerful tool for improving the validity of causal claims based on correlational data¹⁹. Future cross-sectional research should therefore aim to carefully include relevant variables to construct appropriate DAGs that may accurately reflect the underlying causal processes.

Point by Point Response to Reviewer's Comments

Reviewer #1

-

Reviewer #2 (Remarks to the Author):

Reviewer 2 Comment 1. I thank the authors for their constructive engagement with the previous round of feedback. I think the manuscript is much improved, and think it is suitable for publication pending some minor issues that remain.

Response Reviewer 2 Comment 1:

Thank you very much for your thoughtful comments and for assessing the manuscript as substantially improved and suitable for publication. We appreciate the valuable points you raised. We have addressed all remaining minor comments as outlined below.

Reviewer 2 Comment 2. In particular, I appreciate that the authors added nuance regarding trust and its role. However, there are still a few places where the offered claims are more general than is warranted. E.g. p 20 "Study 2 found no evidence that the positive correlation between trust in science and protection intentions reflects a causal link." Again, just adding "self-reported" here would help highlight that participants' responses to this survey need not reflect the more state-like sense of trust that you later mention.

Similarly, on p29: "Another explanation for the absence of effects could be that while our manipulation does change trust in science in general, it may not have changed all relevant dimensions of this trust." Again, I don't think that this manipulation speaks to "trust in science in general", given the trait/state distinction (if both were affected, I'd be willing to see this as "general", but there's no evidence for that). Some mild weakening of the phrasing would be enough to render this unobjectionable.

The remaining comments are mainly aimed at improving flow and clarity.

Response Reviewer 2 Comment 2:

Thank you for pointing out these instances that we overlooked in the previous revision. We have reworded the paragraphs accordingly, removed "general" and added "self-reported," as suggested.

Reviewer 2 Comment 3. *The end of the introduction outlines some key motivations for each study (e.g., fear, fictional future pandemic), but these are not very prominent when each study is later described. E.g., on p12 for study 3, there's no reminder about fear except for a brief mention on p13 that this is a new variable. Which isn't enough to highlight for the reader that this is the whole point of study 3. Similarly, there's no mention of fictional future pandemics on p14 when study 4 is described. Provide a clear reminder for why you ran each study when its design/procedure is explained.*

Response Reviewer 2 Comment 3:

Thank you for highlighting this. In our original manuscript, the Results section appeared before the Methods, and thus we included the motivational introductions as part of the Results (when first introducing each study). As we have now reordered the Methods and Results sections, these introductions no longer appear at the appropriate point in the manuscript (i.e., when the reader first encounters each study). To address your comment, we have thus now moved these motivational introductions to the Methods section, where they now precede the procedural details. In the Results section, we now include only brief reminders of each study's main goal to avoid redundancy.

Reviewer 2 Comment 4. *Figure 1 might benefit from some jittering (or some more transparency in the points) to better show where numerous data points overlap. If there are actually few overlaps, you can ignore this.*

Response Reviewer 2 Comment 4:

Thank you for this comment. We compared the figure with and without jittering and found that jittering indeed better represents all data points. We have therefore replaced the figure with the jittered version and updated the figure legend accordingly.

Reviewer 2 Comment 5. *On p31, you mention a "final possibility". This seems the most compelling possibility to me! But you needn't agree with my assessment. Still, a minor tweak (e.g., "a final, but no less plausible, possibility") could make it clear that this is "last but not least".*

Response Reviewer 2 Comment 5:

Thank you for this suggestion. We have added the phrase “but no less plausible” as recommended.